# Inflammatory Cascade in Alzheimer’s Disease Pathogenesis: A Review of Experimental Findings

**DOI:** 10.3390/cells10102581

**Published:** 2021-09-28

**Authors:** Jade de Oliveira, Ewa Kucharska, Michelle Lima Garcez, Matheus Scarpatto Rodrigues, João Quevedo, Ines Moreno-Gonzalez, Josiane Budni

**Affiliations:** 1Programa de Pós-Graduação em Ciências Biológicas: Bioquímica, Departamento de Bioquímica, Instituto de Ciências Básicas da Saúde, Universidade Federal do Rio Grande do Sul, Porto Alegre 90050-000, Brazil; deoliveirajade10@gmail.com (J.d.O.); matheus_scarodrigues@hotmail.com (M.S.R.); 2Faculty of Education, Institute of Educational Sciences, Jesuit University Ignatianum in Krakow, 31-501 Krakow, Poland; ewa.kucharska@vadimed.com.pl; 3Department of Biochemistry, Federal University of Santa Catarina, Florianópolis 88040-900, Santa Catarina, Brazil; garcez.lima.mi@gmail.com; 4Translational Psychiatry Program, Faillace Department of Psychiatry and Behavioral Sciences, McGovern Medical School, The University of Texas Health Science Center at Houston (UTHealth), Houston, TX 77030, USA; Joao.L.DeQuevedo@uth.tmc.edu; 5Center of Excellence on Mood Disorders, Faillace Department of Psychiatry and Behavioral Sciences, McGovern Medical School, The University of Texas Health Science Center at Houston (UTHealth), Houston, TX 77030, USA; 6Neuroscience Graduate Program, Graduate School of Biomedical Sciences, MD Anderson Cancer Center, UTHealth, The University of Texas Houston, Houston, TX 77030, USA; 7Graduate Program in Health Sciences, Translational Psychiatry Laboratory, University of Southern Santa Catarina (UNESC), Criciuma 88806-000, Brazil; 8Department of Cell Biology, Faculty of Sciences, University of Malaga, IBIMA, 29010 Malaga, Spain; inesmoreno@uma.es; 9Networking Research Center on Neurodegenerative Diseases (CIBERNED), 29010 Malaga, Spain; 10Department of Neurology, McGovern Medical School, The University of Texas Health Science Center at Houston (UTHealth), Houston, TX 77030, USA; 11Programa de Pós-Graduação em Ciências da Saúde, Laboratório de Neurologia Experimental, Universidade do Extremo Sul Catarinense, Criciuma 88806-000, Brazil

**Keywords:** Alzheimer’s disease, neurodegenerative disease, dementia, neuroinflammation, inflammatory cascade, systemic inflammation, BBB disruption, anti-inflammatory effects, therapy

## Abstract

Alzheimer’s disease (AD) is the leading cause of dementia worldwide. Most AD patients develop the disease in late life, named late onset AD (LOAD). Currently, the most recognized explanation for AD pathology is the amyloid cascade hypothesis. It is assumed that amyloid beta (Aβ) aggregation and deposition are critical pathogenic processes in AD, leading to the formation of amyloid plaques, as well as neurofibrillary tangles, neuronal cell death, synaptic degeneration, and dementia. In LOAD, the causes of Aβ accumulation and neuronal loss are not completely clear. Importantly, the blood–brain barrier (BBB) disruption seems to present an essential role in the induction of neuroinflammation and consequent AD development. In addition, we propose that the systemic inflammation triggered by conditions like metabolic diseases or infections are causative factors of BBB disruption, coexistent inflammatory cascade and, ultimately, the neurodegeneration observed in AD. In this regard, the use of anti-inflammatory molecules could be an interesting strategy to treat, delay or even halt AD onset and progression. Herein, we review the inflammatory cascade and underlying mechanisms involved in AD pathogenesis and revise the anti-inflammatory effects of compounds as emerging therapeutic drugs against AD.

## 1. Introduction

An estimated 50 million people worldwide have dementia, a number projected to double every 20 years until 2050 [1,2]. The majority of demented individuals develop Alzheimer’s disease (AD). Thus, AD represents a tremendous healthcare challenge of the 21st century [3,4]. The typical clinical presentation of this age-related neurodegenerative disease is a gradual and progressive decline in different cognitive domains, most commonly involving episodic memory and executive functions that cause social and occupational deficits [5].

AD is characterized by pathological hallmarks such as extracellular amyloid plaques, formed by the deposition of amyloid β-peptide (Aβ), the appearance of intracellular neurofibrillary tangles, composed of hyperphosphorylated tau and extensive synaptic and neuronal loss in the cerebral cortex and hippocampus [6]. In addition, it has been proposed that neuroinflammation is an early feature of AD [7,8]. Thus far, the amyloid cascade hypothesis is the main influential model to explain the progression of AD pathogenesis [9,10]. However, the field is gradually moving away from the simplistic assumption of linear causality proposed in the original amyloid hypothesis [11,12]. Importantly, it has been shown that tau pathogenesis could be associated with neurodegeneration and neuroinflammation regardless of Aβ pathology [13,14]. Moreover, APOE4 is the strongest genetic risk factor for LOAD. The presence of APOE4 is associated with increased Aβ deposition, but also tau pathology [15,16]. In fact, APOE is involved in tau pathogenesis through neuroinflammation processes [17].

AD can be divided into early-onset AD (EOAD) and late-onset AD (LOAD). EOAD is rare, accounting for less than 5% of the cases [18,19] and its onset occurs before 65 years. Mutations in three genes, which encode amyloid protein precursor (APP), presenilin-1 (PSEN1), and presenilin-2 (PSEN2), are known to cause a proportion of autosomal dominant inherent EOAD, or autosomal dominant AD (ADAD). In the genetic form of AD, Aβ accumulation is due to a significant higher peptide production [20]. The most common form, LOAD, is assumed to be a multifactorial and polygenic disease and, therefore, the etiology of Aβ deposition and neurodegeneration in these cases is unknown [19].

Considering the high epidemiological impact of AD, it is fundamental to understand the mechanisms involved in its pathological onset and advancement. In this regard, inflammation seems to play an essential role in disease development and progression. On one hand, clinical and preclinical studies analyzing brains from individuals with AD or experimental models of AD provide evidence for the activation of inflammatory pathways. On the other hand, anti-inflammatory compounds are associated with a reduced risk of developing and disease progression [21,22,23]. During decades, inflammatory processes have been explored in an effort to identify alternative therapeutic targets, alone or in combination with other drugs. In this review, we discuss the most relevant evidences that point out neuroinflammation as a crucial event in AD pathophysiology and its potential as an innovative target to treat AD.

## 2. Pathogenic Mechanisms of Alzheimer’s Disease

Until now, amyloid hypothesis has been the most stablished model of AD pathogenesis. It proposes that the deposition of misfolded and aggregated Aβ is a critical and the initial pathological event in AD, triggering synaptic dysfunction, neuronal loss, and cognitive impairment [10,24].

During the amyloidogenic pathway, Aβ, a peptide of 36–43 amino acids, is generated by cleavage of the transmembrane amyloid precursor protein (APP) through sequential proteolytic processing. In this via, APP is first cleaved by β-secretase (beta-site amyloid precursor protein-cleaving enzyme 1, BACE-1), producing an extracellular soluble fragment (i.e., sβAPP) and an intracellular C-terminal portion termed C99. Subsequently, the resulting cell-associated C-terminal fragments are subjected to intramembrane proteolysis mediated by γ-secretase, which generates a spectrum of Aβ peptides of varied lengths. Concurrently, the non-amyloidogenic APP proteolysis involves cleavage by α- and γ-secretases, resulting in the generation of a long-secreted form of APP (sAPPα) and C-terminal fragments (CTF 83, p3 and AICD50). The APP non-amyloidogenic processing produces non-toxic fragments. The cleavage site for α-secretase in APP lies within the Aβ sequence and thus precludes Aβ formation. Usually, about 90% of APP enters the non-amyloidogenic pathway, while the rest follows the amyloidogenic via [25,26].

The total Aβ burden is regulated by synthesis and clearance rates. In fact, Aβ clearance or degradation, rather than its synthesis, has been considered critical in Aβ accumulation. The clearance of the peptide by transport into the cerebrospinal fluid (CSF), the blood across the blood–brain barrier (BBB), and the removal by macrophages have been suggested as the responsible mechanisms for controlling the brain Aβ levels [27]. Furthermore, CSF clearance seems to be impaired in AD, contributing to increase Aβ burden and disease progression [28]. Some proteases, such as some cathepsins and insulin-degrading enzyme (IDE), play essential roles by cleaving Aβ into shorter soluble fragments without neurotoxic effect [29]. The central receptors for Aβ transport across the BBB from the brain to the bloodstream and from the blood to the brain are low-density lipoprotein receptor (e.g., LRP-1) and the receptor for advanced glycation end products (RAGE), respectively [30]. A chronic imbalance between Aβ production and clearance may result in the agglomeration of intracellular and extracellular aggregates in the brain.

Aβ peptides spontaneously aggregate into soluble oligomers, fibrils, and deposit as senile plaques. These events cause toxicity through several mechanisms, including oxidative injury, microglial and astrocytic activation, as well as altering kinase/phosphatase activity, eventually leading to synaptic damage and neuronal death. It is important to mention that Aβ oligomers are the most neurotoxic form [24,31].

Besides the strong evidence about the relation between Aβ and neurodegeneration, there is a continuous debate about the Aβ hypothesis [10,32]. This is mainly due to the constant failure of developing disease-modifying drugs targeting Aβ, preventing neither its aggregation, accumulation, nor clearance. Nowadays, the Aducanumab efficiency has been extensively discussed, a human IgG1 anti-Aβ monoclonal antibody specific for Aβ oligomers and fibrils [33]. Other reasons are the difficulty correlating the Aβ deposits and AD pathology and the disconnection between Aβ and phosphorylated tau deposition. In fact, tau pathology (tauopathy) has been related to neurodegeneration and neuroinflammation independently of Aβ. In addition, there are substantial differences between familial and sporadic diseases. Importantly, peripheral inflammatory diseases have been considered risk factors for AD development, which may not be directly associated with the Aβ dyshomeostasis [13,32]. This is a discussion that is far from over. Then, new insights into AD pathophysiology are driving the development of drugs towards novel therapeutic targets [34,35]. In this scenario, the inflammatory process has been evaluated as an important component of AD pathogenesis. It is well known that Aβ causes neuroinflammation, and many studies have demonstrated the role of inflammation in the early stages of AD development [36,37].

Moreover, the neurofibrillary tangles (NFT), which Alois Alzheimer first described, are another crucial hallmark of AD pathogenesis [38,39,40]. In this case, the tau protein is aberrantly misfolded and abnormally hyperphosphorylated [40,41]. Tau protein regulates the assembly and stabilization of microtubules. It can be expressed in neurons and oligodendrocytes [42,43,44,45,46].

In AD, the NFT could appear after Aβ accumulation. Considering that plaque-associated dystrophic neurites are not associated with tau, it is probably true that Aβ-mediated neuritic dystrophy occurs first, and the tau accumulation is a consequence of this [47,48,49]. Furthermore, a study conducted by Hurtado et al. [50] showed that Aβ accelerated NFT formation and enhanced tau amyloidosis. Thus, it seems in AD that Aβ plaque deposition drives cortical tau pathology and tau-mediated neurodegeneration [51,52]. In AD neurodegeneration, Aβ diffuse deposits, non-neuritic plaques occur first. Then, the microglia are activated by Aβ deposits, inducing dystrophic neurites that lack tau. This leads to the aggregation of tau hyperphosphorylated facilitating the spread of tau from the limbic system to the cerebral neocortex. The tau hyperphosphorylated distributes from the plaque-associated dystrophic neurites forming NFT throughout the neuron, causing neuronal damage and dementia [53,54,55,56,57].

These events showed the relationship between Aβ and tau, mediated by microglia, causing neurodegeneration and the consequent development of dementia [56].

## 3. Inflammatory Cascade in Alzheimer’s Disease

The innate immune system is the first line of defense against pathogens [58]. The microglia have a fundamental role in early response to central nervous system (CNS) alterations such as damage or infection, development, and homeostasis [59]. Microglia activation is an important event during aging and in neurodegenerative diseases. These cells participate in neuroinflammatory events directly via phagocytosis and cytokine production, as shown by identifying disease-specific microglia, or indirectly responding to cues from the adaptive immune system [60,61].

Neuroinflammation plays a crucial role in the pathogenesis of AD. Several studies have reported the presence of inflammatory markers in the brain of patients, including elevated levels of cytokines/chemokines in serum and CSF, along to microgliosis [62,63,64,65,66]. The increase in these molecules is positively correlated to the cognitive impairment at different stages of AD as well as in individuals with mild cognitive impairment (MCI) [67,68,69].

Recent genome-wide association studies have shown that most polymorphisms recently found in AD patients are involved in the immune response and microglial function. For instance, complement receptor 1 (CR1), CD33, membrane-spanning 4A (MS4A), clusterin (CLU), ATP-binding cassette sub-family A member 7 (ABCA7), sortilin-related receptor 1 (SORL1), inositol polyphosphate-5-phosphatase D (INPP5D), and triggering receptor expressed on myeloid cells 1 and 2 (TREM1, TREM2) [70,71]. Among them, the most prominent polymorphism was found in TREM2, which is associated with phagocytosis [72].

Several studies have shown that Aβ can activate microglia and inflammatory cytokines production [73,74,75]. Even in the prodromal stages of AD, Aβ soluble oligomers can impair synaptic plasticity, inhibit long-term potentiation, and activate microglia [10]. The intracerebral administration of Aβ causes neuroinflammation and memory impairment even in normal adult rodents [74,76,77]. In addition, neuroinflammation and proinflammatory cytokines increase tau phosphorylation and decrease synaptophysin levels, which leads to cytoskeletal instability and neuronal death [78].

The activation of the immune system in the brain occurs through microglial activation of pattern recognition receptors (PRRs), which identify potentially harmful molecules, activating the innate immune system [79]. Several studies have shown that Aβ species can activate PRRs, consequently triggering an immune response [74,80,81,82,83].

Aβ activates several microglial receptors, such as CD36, a class B scavenger receptor, causing secretion of cytokines, chemokines, and reactive oxygen species [84]. Its binding to RAGE induces inflammatory pathways and increases expression of proinflammatory cytokines, such as TNFα and IL-6 [85]. However, the best described pathway is through the activation of Toll-like receptors (TLRs), including TLR2, TLR4, and TLR6 and TREM2 [86,87]. The TLR pathway is responsible for the maturation and release of IL-1β, one of the main pro-inflammatory cytokines involved in the pathophysiology of AD. In fact, IL-1β polymorphism is correlated with the age at AD onset in humans, whereas inhibition of its receptor recues cognitive impairment in animal models [88,89]. IL-1β is produced as a precursor, pro-IL-1β, and requires cleavage to become biologically active. To this end, the pro-IL-1β is cleaved by a complex of intracellular proteins that form the nucleotide-binding oligomerization domain-like receptor family pyrin domain containing 3, known as the NLRP3 inflammasome [90,91]. For NLRP3 assembly, two signals are needed. The first one is the activation of TLRs by Aβ or another potentially harmful molecule. After TLR activation, the signal is transduced through myeloid primary response protein 88 (MyD88), activating nuclear factor kappa B (NF-kB), which leads to transcription of NLRP3 components and proinflammatory cytokines, mainly IL-1β. The second signal is triggered by damage-associated molecular patterns (DAMPs), such as heat shock proteins or ATP, released after cell death. It was proposed by Heneka and col. that this second signal also occurs through phagocytosis of Aβ fragments via TREM2 [90].

The Aβ overload leads to a deficient lysosome degradation of the Aβ fragments, leading to lysosome disruption, cathepsin B release, and the induction of NLRP3 inflammasome assembly [90,92,93]. The inflammasome complex assembly includes the NLRP3 protein, the adapter apoptosis-associated speck-like protein containing a CARD (ASC), and the effector caspase-1. After the NLRP3 complex assembles, caspase-1 cleaves the pro-IL-1β and pro-IL-18, generating the mature form of these cytokines, which activate neutrophils, macrophages, and other microglial cells, amplifying the response inflammatory [94], as represented in Figure 1. This inflammatory pathway has been increasingly described in different neurodegenerative diseases [75,90,91,95,96].

The inflammasome activation finally causes cell death by apoptosis and pyroptosis. Pyroptosis is a form of cell death less organized than apoptosis, triggered by inflammation, as Gasdermin-D-mediated pore formation occurs in the membrane and osmotic lysis, with extravasation of intracellular content [97]. When the NLRP3 inflammasome disassembles, its ASC particles are released from the protein complex, which can activate neighboring microglia, perpetuating the immune response. These particles can also bind to Aβ, contributing to its aggregation [98]. Thus, the efficient degradation of the inflammasome components is a critical step to limit the inflammatory response.

The mechanisms of microglial activation by Aβ depositions have remained not fully elucidated. However, the mechanism known is illustrated in Figure 1. Moreover, Aβ induces the secretion of a variety of additional inflammatory molecules. These molecules include members of the minor compounds of the COX metabolism (prostaglandins), short-lived molecules like nitric oxide (NO), and chemokines [90,99,100,101,102,103].

## 4. TREM2 and Alzheimer’s Pathogenesis

The interest in TREM2 increased after the identification of TREM2 variants as risk factors for AD [104,105]. Individuals bearing the heterozygous mutation (rs75932628) in exon 2 of TREM2, a single nucleotide polymorphism that changes arginine to histidine at position 47 (R47H), produce a four-fold increase in the risk of developing AD [105]. This polymorphism has been validated in neuropathology-confirmed cases and has been shown to increase the risk of sporadic AD as significantly as the ApoE ε4 allele [106]. TREM2 polymorphism causes structural changes in the receptor, leading to a partial loss of its function. However, the role of TREM2 in neurodegeneration and AD remains unclear [107].

TREM2 is a single-pass transmembrane protein whose ligand-binding domain includes an extracellular immunoglobulin-like domain, anchoring the protein to the membrane and contains the intramembranous lysine residue necessary for association with its intracellular membrane adaptor, DAP12. The binding of agonists to TREM2 through the DAP12 protein recruits the cytosolic spleen tyrosine kinase (Syk), which, in turn, activates signaling components including, phosphatidylinositol 3-kinase (PI3K), Akt, mitogen-activated protein kinases (MAPK), and increases intracellular calcium levels. Thus, its activation exerts functions such as cell maturation, survival, proliferation, phagocytosis, and inflammatory regulation [108]. TREM2 ligands include bacteria, bacterial cell components such as lipopolysaccharide (LPS), lipoproteins, such as apolipoprotein A (ApoA), ApoB, ApoE, low-density lipoprotein (LDL), DNA, HSP60 chaperone protein, and Aβ [109].

TREM2 expressed on the cell surface can also undergo proteolysis by α-secretase and γ-secretase [110] to generate soluble TREM2 (sTREM2). The catalytically active components of the γ-secretase complex are PSEN 1 and 2, the same proteins mutated in familial AD and responsible for Aβ processing. Inhibition of γ-secretase leads to accumulation of TREM2 c-terminal fragments (CTFs) on the cell surface, impairing signaling and interfering with normal receptor function [111]. This relationship between TREM and PSEN provides evidence about a functional connection between genetic factors found in AD patients.

One of the main mechanisms of TREM2 is undoubtedly its phagocytic activity. In addition, TREM2 expression increases myeloid cell number in response to inflammation or disease [45], besides regulating the inflammatory responses and the clearance of apoptotic neurons and Aβ. However, it may depend on the activating by ligand and the availability of the TREM2 signaling machinery [112]. The AD-associated TREM2 variant, R47H, reduces receptor binding capacity and, consequently, decreases Aβ phagocytosis [113].

In AD patients, microglial cells have decreased phagocytic capacity and a pro-inflammatory phenotype and morphology [114,115]. Therefore, amyloidosis levels can be altered since phagocytosis is one of the principal mechanisms for Aβ clearance. The increase in the microglial TREM2 expression reduces Aβ1-42 soluble and insoluble forms, the formation of senile plaques, and improves cognitive impairment in AD transgenic mice [116]. Conversely, TREM2 deficiency seems to interrupt the formation of the neuroprotective barrier composed of microglia around the amyloid plaques, responsible for their isolation, increasing neuronal toxicity [117]. However, Aβ overload can cause phagocytosis disruption, cathepsin B release, and NLRP3 assembly, which leads to the amplification of the inflammatory response.

Thus, it is controversial whether phagocytosis and inflammation in AD are beneficial or harmful. Perhaps, in the early stages of the disease, the activation of the immune system and induction of phagocytosis can contribute to Aβ clearance, preventing its toxicity and formation of amyloid deposits. However, chronic inflammation can become detrimental because, if the amyloid load cannot be resolved, it can contribute to the progression of AD [118]. This possible dual role of inflammation in AD progression can be related to microglial function. The microglia are important to scavenging duties. However, it produces reactive oxygen species, secretion of proinflammatory cytokines, or degradation of neuroprotective retinoids when activated. In this case, it may thus unnecessarily put surrounding healthy neurons in danger [119]. Then, the microglia are essential during development and homeostasis, performing critical roles in synaptogenesis and synaptic plasticity. However, in aging and AD, the microglial function is altered, leading the detrimental inflammatory environment [120].

## 5. BBB as a Target of Systemic Inflammation: Importance to Alzheimer’s Disease Development

In the brain, BBB blood vessels present particular properties. BBB essentially regulates the passage of substances and cells between blood and the CNS [121]. Several cell types interact to form and support the BBB, which is now referred to as the “neurovascular unit (NVU)” and is composed of the cerebral endothelial cells, basal lamina, astrocytic foot processes, microglia, and pericytes [122].

BBB disruption is closely associated with several neurological diseases [123]. Evidence has pointed out the alteration of BBB as a trigger to AD pathology [124,125,126,127]. Previously, Ujiie and col. observed that BBB permeability is higher in a 10-month-old transgenic mouse model of AD than in age-matched non-transgenic animals [128]. In fact, the impairment in the BBB is already evident in the AD mouse model at 4 months of age. The BBB leakage seems to occur even before the Aβ deposition and the appearance of other pathological hallmarks of the disease [128]. Corroborating the experimental data, the increased BBB permeability has been demonstrated in early AD individuals [129]. Plasma proteins such as immunoglobulin G (IgG), fibrinogen, and albumin, normally unable to pass the BBB, have been detected around senile plaques in the brain of AD patients [130,131,132]. The presence of peptides derived from hemoglobin and prothrombin in AD brains has been associated with increased leakage of blood. Prothrombin is not produced by the normal brain but shows increased levels in AD brains consistent with leakage across a disrupted BBB [133].

Another important point is that BBB is vital to regulate the brain Aβ metabolism and load, and Aβ deposition could result from an inefficient clearance through BBB [132,134]. Firstly, Shibata et al. [135] and other authors pointed out that brain Aβ is mainly cleared across BBB via LRP-1. Importantly, the LRP-1 content is down-regulated in AD brain [132,135,136,137]. After that, the function of other transporters in the Aβ clearance and AD pathogenesis have been studied, such as RAGE [132,134,138]. On the other hand, Aβ deposition appears to cause BBB damage but is not well evidenced [134]. In addition, it has been demonstrated that BBB endothelial cells respond to truncated tau fragments, ultimately resulting in BBB disruption [139,140]

In parallel, other research groups demonstrated the presence and accumulation of peripheral immune cells and increase in pro-inflammatory cytokines, such as IL-1β, in AD patient’s brain [21,141,142,143,144]. The constituents of innate immunity seem to participate in many processes of the underlying pathological cascade in AD. In addition, compiling studies show that innate immunity is involved in the etiology of LOAD [145,146]. In this regard, increased peripheral inflammation levels can be detected in the early stages of AD [147]. A meta-analysis showed that the blood concentrations of several pro-inflammatory mediators, such as IL-6 and IL-1β, are increased in AD patients [62,148]. In line with this, previous studies showed that inflammatory mediators’ levels are enhanced in the plasma of AD patients 5 years before the clinical onset of dementia compared with age-matched individuals [149,150]. However, it is not well established yet whether brain inflammation in AD subjects is a cause or a consequence of the disease. Although it was previously thought that the CNS was an immune-privileged site, it is now admitted that inflammatory processes occur in response to an injury, infection, or disease and the peripheral immune system can infiltrate into the brain to mediate this response [151,152]. Indeed, the systemic inflammation seems to be the causative factor of BBB disruption and, consequently, neurodegeneration and cognitive dysfunction [153,154,155,156]. Although still debatable, evidence suggests that early or lifelong systemic inflammation triggers long-lasting modulation of CNS immune responses leading to AD development in late life [157].

## 6. Systemic Inflammatory Diseases and the Connection with Alzheimer’s Disease Development

The etiology of sporadic AD is complex and associated with several genetic and behavioral risk factors, in addition to aging [19]. Some of these risk factors related to AD are peripheral diseases, for example, metabolic disorders, such as hypercholesterolemia, diabetes, obesity, and hypertension [158,159,160]. One possible common event between systemic and brain diseases is chronic systemic inflammation [161]. In line with this, an essential feature of the metabolic disorder’s physiopathology is the increased production of pro-inflammatory cytokines [162].

The inflammatory response of the peripheral adipose tissue is an important event of diabetes and obesity [163]. In obese individuals, adipocytes, and resident immune cells of adipose tissue, especially lymphocytes and macrophages, contribute to the increased levels of circulating cytokines, such as TNF-α, IL-6, and IL-1β, as well as C reactive protein (CRP) [164]. Nowadays, white adipose tissue (WAT) is known to be a secretory tissue, which seems crucial in brain dysfunction development [165]. In hyperglycemia conditions, the NF-κB—a transcription factor that regulates the induction of several inflammatory genes—is rapidly and strongly activated in vascular cells, resulting in enhanced leukocytes adhesion and pro-inflammatory cytokines transcription [166]. Hyperlipidemias, primarily hypercholesterolemia and hypertriglyceridemia, are also related to systemic inflammation. For instance, the study of Lohmann et al. [167] demonstrated that mice fed with a high cholesterol diet presented generalized inflammation, characterized by increased in T lymphocytes and macrophage recruitment in adipose tissue, inducing cytokine production [167].

Most of these metabolic risk factors are associated with BBB leakage and neuroinflammation. An epidemiological study showed that overweight or obesity in middle-aged individuals is associated with loss of BBB integrity several years later [168]. Diabetes is usually associated with macro- and microvascular complications, including CNS alterations that result, at least in part, in BBB damage. Impairment of the cerebral microvascular structure, characterized by reduction in capillary density and tight junctions damage, is a relevant mechanism of BBB dysfunction induced by diabetes [169]. Our group have recently demonstrated that hypercholesterolemic mice present high BBB permeability in the hippocampus, associated with intense astrogliosis [170,171].

According to experimental and clinical studies, the BBB is altered in hypercholesterolemia, resulting in the infiltration of immune cells in the brain parenchyma and consequently the production of inflammatory mediators [172,173,174]. On one hand, this inflammatory response, associated with persistent activation of glial cells, induces neuronal damage and, ultimately, leads to cognitive dysfunction and dementia [170,171,175]. On the other hand, when the BBB is damaged, the transport of Aβ is defective. RAGE is overexpressed and the expression of LRPs decreases, leading to the accumulation of Aβ in the brain [174,176].

In obese individuals, there is a chronic systemic inflammation that induces production of pro-inflammatory cytokines, such as IL-6, and adipokines (such as TNF-α), mainly produced by WAT. These inflammatory molecules could cause alterations in BBB permeability and, consequently, brain inflammation and neurodegeneration. Deregulation of these molecules could link obesity and AD development [165]. The effects of saturated fatty acids (e.g., palmitic acid) on microglia and astrocyte activation have also been demonstrated. For instance, these fatty acids promote the pro-inflammatory phenotype of microglia, resulting in activation of NF-κB pathway, TLR-4 receptors, interferon-γ (IFN-γ), and TNF-α production [177].

Metabolic diseases are not the only example of systemic inflammatory pathologies associated with BBB disruption and neurodegeneration. Sepsis is another relevant condition that has been implicated in dysfunction and loss of neurons [178,179]. A possible consequence of sepsis is the septic encephalopathy, which occurs in 8–70% of septic patients. It is related to BBB disruption, leucocyte infiltration, up-regulation of aquaporin-4, activation of microglia, astrocytosis, and neuronal death [178]. This relationship between sepsis and brain alterations is an opportunity to better understand the effects of systemic inflammation in cerebral function [179,180,181]. In fact, it has been reported that RAGE may be involved in sepsis-mediated increase in amyloidogenic proteins and cognitive impairment [182].

In this context, viral infections associated with intense systemic inflammation have been a concern in the neuroscience field. For instance, COVID-19 could lead to neurodegeneration and increase the risk of AD due to an intensive brain inflammatory process as a result of systemic inflammation. Furthermore, SARS-CoV-2 (severe acute respiratory syndrome coronavirus 2) is potentially neuroinvasive, suggesting that neurological consequences could occur after direct brain infection [183,184]. Recent studies have reported neurological complications in COVID-19 patients [185,186,187,188].

However, it is important to mention that although, growing evidence indicated the relationship between infections and AD, more studies are needed to elucidate better the mechanisms involved. Notably, chronic viral, bacterial, and other infections might be causative factors for the BBB breakdown and coexistent brain inflammatory pathway activation and consequently neurodegeneration [189,190].

Additionally, gut microbiota alterations can cause neuroinflammation and consequently interfere in AD development. Several reports have pointed out the gut microbiota as a modulator of the neuroinflammatory process in AD [191,192]. Importantly, an imbalance in the gut microbiome is related to systemic inflammation and peripheral conditions like diabetes. Moreover, dietary changes may induce a loss of microbiota ecosystem homeostasis [193,194]. Chronic dysbiosis appears to cause BBB leakage and release of pro-inflammatory molecules, endotoxins, ultimately leading to microglia and astrocytes activation [195]. This scenario also increases intestinal permeability, promoting translocation of bacteria and endotoxins across the epithelial barrier and activation of both enteric neurons and glial cells [191,196]. Moreover, the oral microbiota are also related to AD [197,198]. For instance, periodontitis, the most common chronic oral bacterial infection in adults, is generally caused by *Porphyromonas gingivalis* [199]. This bacterium has been detected in the brains of AD patients [197], indicating a strong association between periodontal pathogens and AD [198]. Additionally, the reduction in GSK3β activation may help delay the periodontitis-promoted pathological progression of AD [200]. Figure 2 represents some events connecting systemic inflammatory conditions and AD development.

## 7. Anti-Inflammatory Approaches in the Alzheimer’s Disease

Current approaches for AD management, based on neurotransmission dysfunctions, are not sufficient. These therapies, such as acetylcholinesterase inhibitors and memantine, do not modify the natural course and outcome of the disease. Available treatments are palliative rather than curative or disease-modifying therapies [201]. In this regard, a series of anti-inflammatory drugs have been pointed out as therapeutic strategies to control AD progression [202].

Epidemiological and experimental studies suggest a positive effect of the treatment with non-steroidal anti-inflammatory drugs (NSAIDs) in AD [203,204,205,206,207,208]. Earlier experimental works indicated that BACE1 expression (mRNA and protein) is stimulated by pro-inflammatory mediators and inhibited by NSAIDs [209,210]. Other studies indicated that treatment with certain NSAIDs decreased brain Aβ accumulation in animal models of AD, which was related to anti-inflammatory mechanisms [211,212,213]. In line with this, in a preclinical study, Medeiros and col. [214] reported that aspirin, the most commonly used NSAID, decreased activation of NF-κB and generation of pro-inflammatory molecules in Tg2576 mice. These anti-inflammatory effects caused the activation of phagocytic microglia, resulting in Aβ clearance and improvement of memory. However, these compounds, considered classic anti-inflammatory molecules, have not convincingly shown any beneficial effects during clinical trials in AD patients [215]. In fact, the actions of NSAIDs in dementia depend on the stage of disease progression. Lichtenstein and col. [216] affirmed that the motto for NSAID therapeutics in AD should be “the earlier, the better”.

Other studies have examined different anti-inflammatory drugs, e.g., glucocorticoids. Minocycline, for instance, reduces inflammatory parameters in the brain and serum and reverses memory impairment in a mouse model of AD [73]. Specifically, minocycline administration reduces the production of IL-1β, TNF-α, IL-4, and IL-10 induced by Aβ42 oligomer inoculation in the hippocampus and cortex of mice, which was associated with an improvement in spatial memory. Moreover, the anti-inflammatory effects of minocycline exposure involve TLR2 receptors and NLRP3 [75].

Natural products (e.g., nicotine, vitamin D, vitamin E, melatonin, and resveratrol) also present promising effects in AD [217,218]. Experimental evidence demonstrates anti-inflammatory effects and decreased Aβ levels induced by natural compounds [219,220,221,222,223]. For instance, high vitamin D3 diet caused a decrease in amyloid plaques in the brain of APP transgenic mice. The effect of the vitamin D3 supplementation was correlated with diminished levels of TNF-α in the brains of the AD transgenic mouse model [220]. Zhao and col. [221] showed that resveratrol administration in an experimental model of AD (female rats ovariectomized treated with galactose) decreased the NF-κBp65 and RAGE expression and increased the expression of claudin-5 in the hippocampus. The control induced by resveratrol exposure on neuroinflammation, and BBB permeability reduced the content of insoluble Aβ1–42 in the hippocampus of the rats. Omega-3 fatty acids (e.g., α-linolenic acid) can regulate the microglial activation and control brain inflammation, which seems to prevent AD [177,224,225]. Supplementation with oil fish, containing eicosapentaenoic acid and docosahexaenoic acid, decreased neuroinflammation and improved cognitive impairment in septic aged rats [226].

However, when some anti-inflammatory agents are tested in AD patients, the results are not satisfactory. For instance, Nivaldipine that showed potential anti-inflammatory effects in animals, was not beneficial in treating mild to moderate AD [227]. For cognitively intact individuals, low-dose naproxen does not significantly reduce the progression of presymptomatic AD [228]. A clinical trial showed that minocycline did not delay cognitive or functional impairment progress in patients with mild AD over 2 years [229]. Other clinical trials performed with aspirin showed no evidence of reducing the risk of dementia, MCI, or cognitive decline [230].

Nanotechnology has also been tested as an anti-inflammatory strategy to treat neuropathologies, particularly in AD [231,232]. Gold nanoparticles treatment prevented neuroinflammation and cognitive impairment in a rat model of sporadic dementia [231]. Rats exposed to streptozotocin, a sporadic AD model, that present memory impairment, increased levels of IL-1β and NF-κB, and showed to improve after gold nanoparticle treatment. Additionally, gold nanoparticle administration ameliorated BBB disruption and brain dysfunction in hypercholesterolemic mice by improving peripheral inflammation [233]. However, more studies are needed, especially to better establish the safety of gold nanoparticle administration. Figure 3 summarizes the main mechanisms of anti-inflammatory drugs and compounds that display anti-inflammatory effects in AD brains.

## 8. Conclusions

AD is a complex, multifactorial, heterogeneous, and severe neurodegenerative disease. It initiates many years before symptoms, as illustrated in Figure 4. Many risk factors are responsible for the development of AD, including genetics, aging, bacteria, diabetes, gut dysbiosis, hypercholesterolemia, obesity, and virus. These risk factors induced systemic inflammation (1) and BBB disruption (2). Thus, Aβ aggregation, tau hyperphosphorylation and, neuroinflammation occur as a possible consequence (3). The neuroinflammation involves the glial activation and release of inflammatory mediators such as NO, IL1- β, Il-18, TNF-α, prostaglandins, and chemokines such as fractalkine (CX3CL1), MIP-1α (CCI3), IP10 (CXCL10) and, MCP-1 (CCL2). These events lead to neuronal death (4) that culminates in memory loss and changes in mood or personality, featuring dementia-like DA (5). Therefore, anti-inflammatory drugs and compounds that display anti-inflammatory effects in AD brains can be an interesting strategy for AD. Finally, it is essential to mention that due to the existence of many failed pathways involved with AD pathogenesis, the success of anti-inflammatory therapy to treat or prevent AD is still impaired. This phenomenon is exemplified in clinical trials with anti-inflammatory molecules. Therefore, just one anti-inflammatory agent does not have benefit enough in the disease. A combination of therapeutic agents may be needed to have the most significant potential to prevent or treat AD development and/or progression.

## Figures and Tables

**Figure 1 cells-10-02581-f001:**
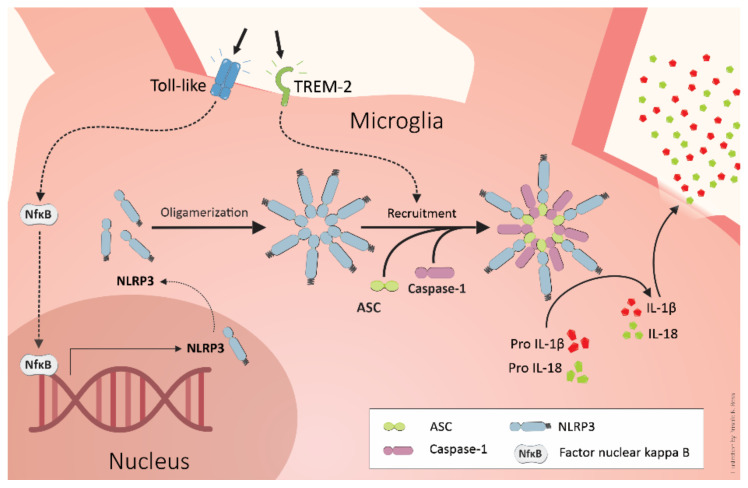
The activation of Toll-like receptors by ligands such as LPS or Aβ species can trigger the first signal for assembly of the NLRP3 inflammasome through nuclear factor kappa B (NF-kB). The second signal can be triggered after Aβ-mediated phagocytosis by TREM2. The Aβ overload causes lysosome disruption, releasing cathepsin B, which induces the signal for assembly of NLRP3. The inflammasome contains caspase-1 that cleaves pro-IL-1β and pro-IL-18, generating the mature forms of these inflammatory cytokines. IL-1β and IL-18, in turn, activate neutrophils, macrophages, and other microglial cells, amplifying the inflammatory response.

**Figure 2 cells-10-02581-f002:**
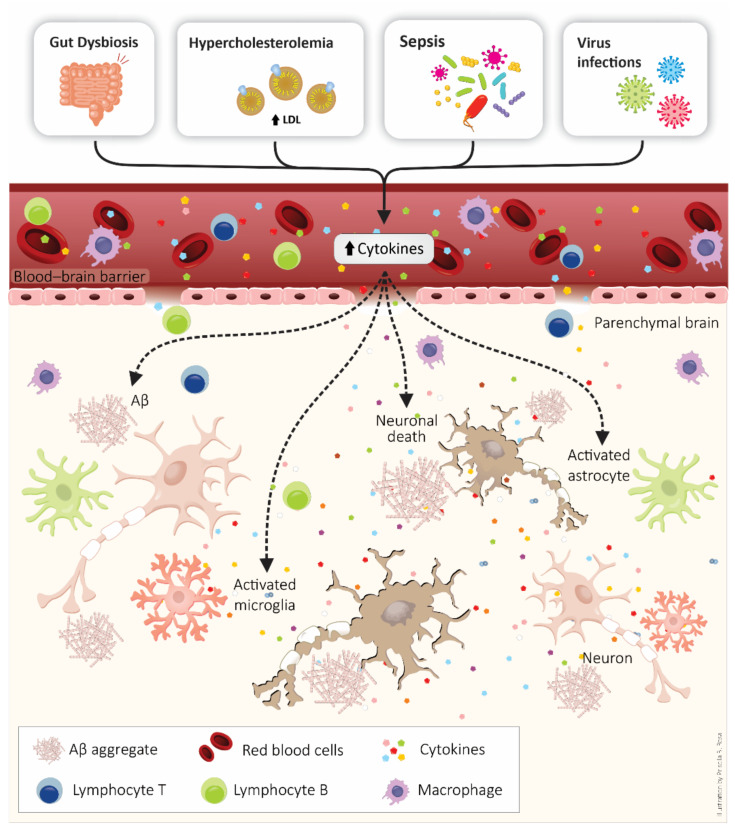
Peripheral diseases are a risk factor for β-amyloid peptide (Aβ) peptide accumulation, neurodegeneration, and Alzheimer’s disease development. Systemic inflammatory conditions, such as metabolic disease, sepsis, virus infections, and dysbiosis, are associated with blood–brain barrier (BBB) disruption and coexistent neuroinflammation. Neuroinflammation is characterized by the presence of the peripheral immune system, activation of glial cells (astrocytes and microglia), and increased production of pro-inflammatory molecules (e.g., cytokines).

**Figure 3 cells-10-02581-f003:**
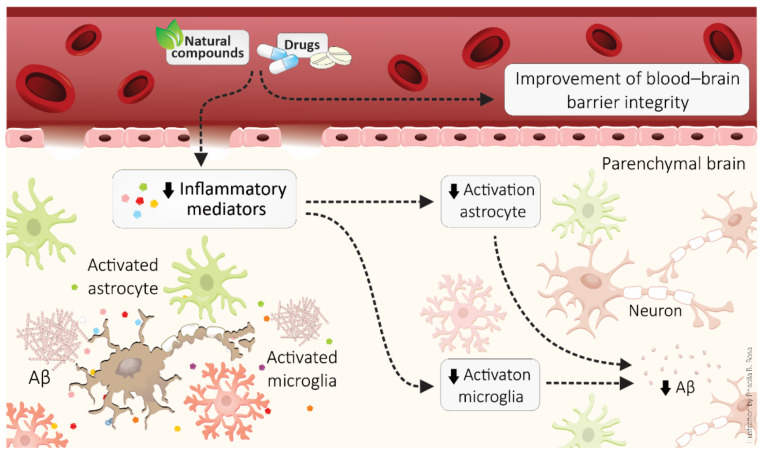
Anti-inflammatory approaches, such as drugs and natural compounds, in Alzheimer’s disease. Anti-inflammatory therapeutic strategies improve the blood–brain barrier and neuroinflammation, decreasing activation of astrocytes and microglia as well as generation of pro-inflammatory molecules.

**Figure 4 cells-10-02581-f004:**
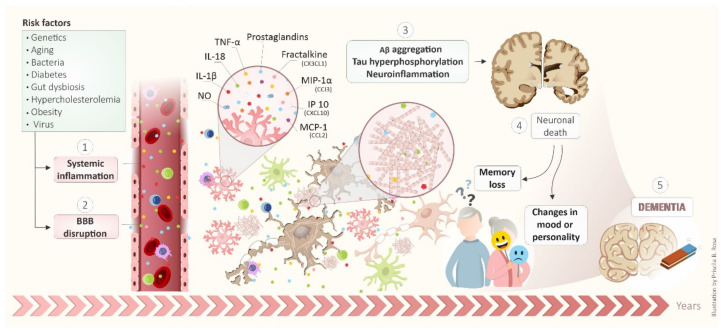
Schematic integrated view of mechanisms involved in development and progression of AD.

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
