# Peer review of "Inflammatory Cascade in Alzheimer’s Disease Pathogenesis: A Review of Experimental Findings"

_cells, 2021, doi:10.3390/cells10102581_

Round 1

Reviewer 1 Report

In this Review, the Authors go inside the importance of inflammation for Alzheimer's Disease (AD) pathogenesis. The topic is of current interest and I have some suggestions to imprtove specific points in the text.

Major points

1) Introduction.

a) Please add some references about the role of Tau protein in AD inflammation. For instance, start from:

Tau and neuroinflammation: What impact for Alzheimer's Disease and Tauopathies? Laurent C, Buée L, Blum D.Biomed J. 2018 Feb;41(1):21-33. doi: 10.1016/j.bj.2018.01.003. Epub 2018 Mar 20.

b) Line4s 67-68. The debate around the pathogenic role of Abeta is becoming more and more relevant, I suggest to expand this topic that here is presented and briefly deepened afterwards (lines 124-130). More details about the reasons for Abeta hypothesis confutation may be useful to the reader also to appreciate the importance of alternative approaches focusing on peripheral inflammation.

2) Pathogenic mechanisms of Alzheimer’s disease

Please add some references addressing the role of tau and how it is considered in relationship to Abeta from the pathogernic point of view.

3) TREM2 and Alzheimer´s pathogenesis

Lines 263-268. As this is a very important point, please comment more extensively and add proper references about the possible dual role of inflammation in AD progression.

4) BBB as a target of systemic inflammation: importance to Alzheimer’s disease development

Lines 305-307. Please add a couple of possible mechanisms linking BBB disruption to Abeta and tau aggregation and neurotoxicity to reinforce the importance of BBB leakage in AD pathogenesis.

5) Systemic inflammatory diseases and the connection with Alzheimer’s disease development

a) I would suggest to be cautious in associating inflammation coming from acute infectious disease as trigger for AD. In fact, the result of encephalitis may be neurodegeneration but this is an atypical form (see for instance:

Autoimmune Encephalitis Resembling Dementia Syndromes.

Bastiaansen AEM, van Steenhoven RW, de Bruijn MAAM, Crijnen YS, van Sonderen A, van Coevorden-Hameete MH, Nühn MM, Verbeek MM, Schreurs MWJ, Sillevis Smitt PAE, de Vries JM, Jan de Jong F, Titulaer MJ.Neurol Neuroimmunol Neuroinflamm. 2021 Aug 2;8(5):e1039. doi: 10.1212/NXI.0000000000001039. Print 2021 Sep.  

Pleasde evidence this point discussing more deeply how infectious disorders can lead to chronic neurodegeneration (and not to acute, rapid neurodegenetation).

b) I would also suggest to add in the microbiota paragraph this reference (or others) that underscores a possible direct link between microbial invasion and AD (from the oral microbiota, so, not just guit microbiota is relevant):

GSK3beta is involved in promoting Alzheimer's disease pathologies following chronic systemic exposure to Porphyromonas gingivalis lipopolysaccharide in amyloid precursor protein(NL-F/NL-F) knock-in mice.

Jiang M, Zhang X, Yan X, Mizutani S, Kashiwazaki H, Ni J, Wu Z.Brain Behav Immun. 2021 Aug 13;98:1-12. doi: 10.1016/j.bbi.2021.08.213.

6) Anti-inflammatory approaches in the Alzheimer’s disease

a) This paragraph presents a quick overview of the topic reported in its title. I would suggest to include some sentences summarizing with few examples current trends in clinical trials ongoing in AD using molecules having a supposed antioinflammatory effect. This is important to appreciate how the clinical field is moving to this respect.  

b) Lines 437-445. The presented papers with gold nanoparticles are interesting, but nanoparticles are also concerning because of their side effects (see for instance: 

Electrophysiological effects of polyethylene glycol modified gold nanoparticles on mouse hippocampal neurons. Tuna BG, Yesilay G, Yavuz Y, Yilmaz B, Culha M, Maharramov A, Dogan S.Heliyon. 2020 Dec 28;6(12):e05824. doi: 10.1016/j.heliyon.2020.e05824. eCollection 2020 Dec.PMID: 33426332   

I would suggest to avoid conveying the message that this may be (as it is) a promising therapy for neuroinflammation.  

7) Conclusions

I would add "genetics" to the AD risk factor list. I also suggest that the Authors summarize here, as take home message, which main factors currently prevent the success of antinflammatory strategies in AD therapy (or prevention?)

Author Response

Response to Reviewer 1 Comments

Reviewer#1

In this Review, the Authors go inside the importance of inflammation for Alzheimer's Disease (AD) pathogenesis. The topic is of current interest and I have some suggestions to imprtove specific points in the text.

Major points

1) Introduction.

  1. a) Please add some references about the role of Tau protein in AD inflammation. For instance, start from:

Tau and neuroinflammation: What impact for Alzheimer's Disease and Tauopathies? Laurent C, Buée L, Blum D.Biomed J. 2018 Feb;41(1):21-33. doi: 10.1016/j.bj.2018.01.003. Epub 2018 Mar 20.

  1. b) Line4s 67-68. The debate around the pathogenic role of Abeta is becoming more and more relevant, I suggest to expand this topic that here is presented and briefly deepened afterwards (lines 124-130). More details about the reasons for Abeta hypothesis confutation may be useful to the reader also to appreciate the importance of alternative approaches focusing on peripheral inflammation.

Reply: We would like to thank Reviewer #1 for his/her comments. As suggested by the Reviewer, we now added more references regarding the role of tau in Alzheimer’s disease pathogenesis and the Abeta hypothesis in the Introduction (Lines 70-75).

2) Pathogenic mechanisms of Alzheimer’s disease

Please add some references addressing the role of tau and how it is considered in relationship to Abeta from the pathogernic point of view.

Reply: As the Reviewer request, we provided information about tau together with the other pathogenic mechanisms of Alzheimer’s disease (Lines 150-168).

3) TREM2 and Alzheimer´s pathogenesis

Lines 263-268. As this is a very important point, please comment more extensively and add proper references about the possible dual role of inflammation in AD progression.

Reply: Thank you for this valuable comment. We now commented and added more proper references about the dual role of inflammation in AD progression (Lines 299-311).

4) BBB as a target of systemic inflammation: importance to Alzheimer’s disease development

Lines 305-307. Please add a couple of possible mechanisms linking BBB disruption to Abeta and tau aggregation and neurotoxicity to reinforce the importance of BBB leakage in AD pathogenesis.

Reply: We apologize for that omission. We now added more details about the BBB disruption, Abeta and tau aggregation, and neurotoxicity in the Manuscript (Lines 334-342).

5) Systemic inflammatory diseases and the connection with Alzheimer’s disease development

  1. a) I would suggest to be cautious in associating inflammation coming from acute infectious disease as trigger for AD. In fact, the result of encephalitis may be neurodegeneration but this is an atypical form (see for instance:

Autoimmune Encephalitis Resembling Dementia Syndromes.

Bastiaansen AEM, van Steenhoven RW, de Bruijn MAAM, Crijnen YS, van Sonderen A, van Coevorden-Hameete MH, Nühn MM, Verbeek MM, Schreurs MWJ, Sillevis Smitt PAE, de Vries JM, Jan de Jong F, Titulaer MJ.Neurol Neuroimmunol Neuroinflamm. 2021 Aug 2;8(5):e1039. doi: 10.1212/NXI.0000000000001039. Print 2021 Sep.  

Pleasde evidence this point discussing more deeply how infectious disorders can lead to chronic neurodegeneration (and not to acute, rapid neurodegenetation).

Reply: We apologize for that omission. We now added the information more cautiously.

  1. b) I would also suggest to add in the microbiota paragraph this reference (or others) that underscores a possible direct link between microbial invasion and AD (from the oral microbiota, so, not just guit microbiota is relevant):

GSK3beta is involved in promoting Alzheimer's disease pathologies following chronic systemic exposure to Porphyromonas gingivalis lipopolysaccharide in amyloid precursor protein(NL-F/NL-F) knock-in mice.

Jiang M, Zhang X, Yan X, Mizutani S, Kashiwazaki H, Ni J, Wu Z.Brain Behav Immun. 2021 Aug 13;98:1-12. doi: 10.1016/j.bbi.2021.08.213.

Reply: As suggested by the Reviewer, the reference was added in the new version of the Manuscript (Lines 443-448).

6) Anti-inflammatory approaches in the Alzheimer’s disease

  1. a) This paragraph presents a quick overview of the topic reported in its title. I would suggest to include some sentences summarizing with few examples current trends in clinical trials ongoing in AD using molecules having a supposed antioinflammatory effect. This is important to appreciate how the clinical field is moving to this respect.

Reply: As suggested by the Reviewer, we now added information about the clinical trials of AD patients using anti-inflammatory compounds (Lines 502- 509).

  1. b) Lines 437-445. The presented papers with gold nanoparticles are interesting, but nanoparticles are also concerning because of their side effects (see for instance: 

Electrophysiological effects of polyethylene glycol modified gold nanoparticles on mouse hippocampal neurons. Tuna BG, Yesilay G, Yavuz Y, Yilmaz B, Culha M, Maharramov A, Dogan S.Heliyon. 2020 Dec 28;6(12):e05824. doi: 10.1016/j.heliyon.2020.e05824. eCollection 2020 Dec.PMID: 33426332   

I would suggest to avoid conveying the message that this may be (as it is) a promising therapy for neuroinflammation.  

Reply: As suggested by the Reviewer, we provided a sentence mentioning that more studies are needed to study the safety of gold nanoparticles administration.

7) Conclusions

I would add "genetics" to the AD risk factor list. I also suggest that the Authors summarize here, as take home message, which main factors currently prevent the success of antinflammatory strategies in AD therapy (or prevention?)

Reply: The observations made by the Reviewer are very pertinent. According to the Reviewer's request, we added genetic as a risk factor for Alzheimer's disease. In addition, we included a possible reason that interferes with the success of anti-inflammatory strategies in Alzheimer's disease therapy.

Reviewer 2 Report

At line 84-86, the authors report “We propose that inflammatory processes have to be explored deeper to be considered as an essential therapeutic target in combination with other drugs.”. The proposal to better explore inflammation in Alzheimer's is not a their idea, but it is a topic that has been proposed, has been known and has been debated for at least 20 years (Inflammation and Alzheimer’s disease, Neuroinflammation Working Group, Neurobiol Aging. 2000 May-Jun; 21(3): 383–421).

It is also necessary to insert more up-to-date references, because most refer to 2017 and only some refer to recent years. Just as an example of the last few months: doi: 10.1080/10717544.2021.1937383, doi: 10.1007/s12035-021-02460-4.

In conclusion, a severe restyling of the bibliographic research is needed.

Author Response

Response to Reviewer 2 Comments

Reviewer#2

At line 84-86, the authors report “We propose that inflammatory processes have to be explored deeper to be considered as an essential therapeutic target in combination with other drugs.”. The proposal to better explore inflammation in Alzheimer's is not a their idea, but it is a topic that has been proposed, has been known and has been debated for at least 20 years (Inflammation and Alzheimer’s disease, Neuroinflammation Working Group, Neurobiol Aging. 2000 May-Jun; 21(3): 383–421).

Reply: We would like to thank Reviewer #2 for his/her comments. The observation made by the Reviewer is very pertinent. As the Reviewer can see, we corrected this information in the Manuscript.

It is also necessary to insert more up-to-date references, because most refer to 2017 and only some refer to recent years. Just as an example of the last few months: doi: 10.1080/10717544.2021.1937383, doi: 10.1007/s12035-021-02460-4.

In conclusion, a severe restyling of the bibliographic research is needed.

Reply:  We apologize for overlooking the references. As suggested by the Reviewer, we added new references in the Manuscript.

Round 2

Reviewer 1 Report

In this revised version, the Manuscript has improved. I have just minor points.

Minor point

1) The following sentence should be revised as it seems blunt.

"This review discusses the main facts that point inflammatory cascade as a crucial event in AD pathophysiology. For many years, inflammatory processes have been explored deeper and considered an important therapeutic target combined with other drugs... (?)

2) Please revise again the added parts (in yellow) for English and typos.

Reviewer 2 Report

That's fine